# Identification and Validation of Ikaros (*IKZF1*) as a Cancer Driver Gene for Marek’s Disease Virus-Induced Lymphomas

**DOI:** 10.3390/microorganisms10020401

**Published:** 2022-02-09

**Authors:** Alec Steep, Evin Hildebrandt, Hongen Xu, Cari Hearn, Dmitrij Frishman, Masahiro Niikura, John R. Dunn, Taejoong Kim, Steven J. Conrad, William M. Muir, Hans H. Cheng

**Affiliations:** 1Genetics Program, Michigan State University, East Lansing, MI 48824, USA; alec.steep@gmail.com; 2Avian Disease and Oncology Laboratory, US National Poultry Research Center, Agricultural Research Service, USDA, East Lansing, MI 48823, USA; evin.hildebrandt@gmail.com (E.H.); cari.hearn@usda.gov (C.H.); 3Genome-Oriented Bioinformatics, Technical University of Munich, D-80333 Munich, Germany; hongen_xu@hotmail.com (H.X.); d.frishman@wzw.tum.de (D.F.); 4Faculty of Health Sciences, Simon Fraser University, Burnaby, BC V5A 1S6, Canada; masahiro_niikura@sfu.ca; 5Endemic Poultry Viral Diseases Research, US National Poultry Research Center, Agricultural Research Service, USDA, Athens, GA 30605, USA; john.dunn@usda.gov (J.R.D.); taejoong.kim@usda.gov (T.K.); steven.conrad@usda.gov (S.J.C.); 6Department of Animal Sciences, Purdue University, West Lafayette, IN 47907, USA; wmmuir@icloud.com; 7Genesys Bioinformatics Service, Punta Gorda, FL 33950, USA

**Keywords:** chicken, Marek’s disease, cancer driver gene, Ikaros, somatic mutation, Meq, recombinant virus

## Abstract

Marek’s disease virus (MDV) is the causative agent for Marek’s disease (MD), which is characterized by T-cell lymphomas in chickens. While the viral Meq oncogene is necessary for transformation, it is insufficient, as not every bird infected with virulent MDV goes on to develop a gross tumor. Thus, we postulated that the chicken genome contains cancer driver genes; i.e., ones with somatic mutations that promote tumors, as is the case for most human cancers. To test this hypothesis, MD tumors and matching control tissues were sequenced. Using a custom bioinformatics pipeline, 9 of the 22 tumors analyzed contained one or more somatic mutation in Ikaros (*IKFZ1*), a transcription factor that acts as the master regulator of lymphocyte development. The mutations found were in key Zn-finger DNA-binding domains that also commonly occur in human cancers such as B-cell acute lymphoblastic leukemia (B-ALL). To validate that *IKFZ1* was a cancer driver gene, recombinant MDVs that expressed either wild-type or a mutated Ikaros allele were used to infect chickens. As predicted, birds infected with MDV expressing the mutant Ikaros allele had high tumor incidences (~90%), while there were only a few minute tumors (~12%) produced in birds infected with the virus expressing wild-type Ikaros. Thus, in addition to Meq, key somatic mutations in Ikaros or other potential cancer driver genes in the chicken genome are necessary for MDV to induce lymphomas.

## 1. Introduction

Poultry is both the most consumed and fastest-growing meat per capita worldwide [1]. Mainly due to advanced poultry breeding, tremendous progress in production traits has been made to meet the rapidly growing demands of consumers for poultry meat and eggs. While highly successful, the poultry industry will need to address several major issues soon. With high-density chicken rearing, reduced genetic diversity from industry consolidation [2], and limitations on antibiotic usage driven by consumers and regulations, controlling infectious diseases and preventing disease outbreaks are critical for sustaining economic viability, maintaining public confidence in poultry products, and enhancing animal welfare.

Among poultry diseases, Marek’s disease (MD), a lymphoproliferative disease caused by the highly oncogenic alphaherpesvirus Marek’s disease virus (MDV, aka *Gallid alphaherpesvirus 2*), is regularly listed as an issue affecting poultry producers. Concern regarding MD is heightened due to the unpredictable, yet recurrent vaccine breaks that result in high losses to poultry farms. Estimated worldwide annual losses from MD due to vaccination costs, meat condemnation, and reduced egg production exceed USD 2 billion [3].

As virulent MDV is ubiquitous in the environment, the main MD control strategy for commercial chickens is widespread vaccination. Unfortunately, while effective in preventing tumors, MD vaccines do not prevent MDV infection or shedding of pathogenic MDV [4]. Considering vaccine viruses and pathogenic MDVs coexist in MD-vaccinated flocks, it is likely that vaccination programs have promoted the evolution of MDV with higher virulence [5,6,7,8]. Specifically, it has been hypothesized that MD-vaccinated flocks select for MDV strains that replicate and/or spread better, which in turn promotes evolution to higher virulence, as this increased viral load favors the likelihood of more transformed cells. Based on pathogenicity shifts, a new MD vaccine is effective for about 10–20 years [9]. With the United States’ introduction of the Rispens strain vaccine (CVI988) in the 1990s and no new vaccine with substantially superior protective efficiency [10], there is growing fear that another major MD outbreak will occur in the near future.

In order to achieve sustainable MD control, future rationally designed approaches will likely require a complete understanding of MDV-induced transformation. A key previous finding was the identification of Meq, a bZIP transcription factor, as the viral oncogene [11]. Additional experiments in which *Meq* was deleted from the MDV genome validated that Meq is necessary for transformation [12]. More recently, recombinant viruses with alleles from MDVs that vary in pathogenicity have clearly demonstrated that key *Meq* polymorphisms are associated with virulence and evading vaccinal resistance [13]. However, other viral genes in the ~175 Kb genome are also likely to be relevant. For example, recent association studies to identify additional MDV genes associated with virulence identified a number of viral genes beyond *Meq*, which is consistent with the belief that MDV-induced pathogenesis is a complex trait [14,15].

Unlike most other herpesviruses, MDV genomes integrate into the host genome as part of their natural lifecycle, and MDV genomes are found in all MD tumors [16]. Integration of MDV occurs near the telomeres, which is facilitated by the presence of repetitive sequences in the viral genome that are identical to host telomeric repeat sequences (TTAGGG) (reviewed by [17]). However, MDV integration(s) into the host genome are insufficient to induce tumor formation, as integration events are frequent (though variable with respect to chromosomal location) early after infection in bursal, thymic, and spleen-derived T cells [18,19]. Furthermore, certain profiles are favored in tumor lineages and tumors are most often clonal based on T-cell receptor (TCR) spectratyping within a single bird, suggesting a strong restriction or selection process of the transformed cell [18,19]. These results, along with the extensive biomedical literature, strongly suggest that additional somatic alterations (such as those in genes that cooperate in or enhance Meq-driven regulation) are necessary to generate MD tumors that progress to gross lymphomas.

In the cancer biology field, a major focus has been to identify and characterize what are known as cancer driver genes. Specifically, these are genes in which polymorphisms or somatic mutations promote tumor growth. More familiar names for cancer driver genes are oncogenes or tumor-suppressor genes, which, in general, promote or inhibit tumor formation, respectively. When altered from the normally functioning allele, these genes will influence one or more of the six hallmarks of cancer pathways: sustaining proliferative signaling, evading growth suppressors, activating invasion and metastasis, enabling replicative immortality, inducing angiogenesis, and resisting cell death [20]. Greatly aided by next-generation sequencing platforms, thus far, 568 cancer driver genes have been cataloged in 66 different cancer types [21].

Given that every susceptible bird infected with virulent MDV does not develop a tumor, we hypothesized that the chicken genome contained cancer driver genes that likely work in conjunction with Meq. Sequencing DNAs from MD tumors and their matching controls, somatic mutations were repeatedly identified in key domains of Ikaros (*IKZF1*) that are also known to exist in several human cancers. Validation that Ikaros acts a MD cancer drive gene was supported with the use of recombinant MDVs that expressed either wild-type or a mutant Ikaros allele found in several MD tumors. These results provided considerable biological insights on MDV-induced transformation that should aid in the rational design of more efficient or novel MD control measures, as well as broader implications for the importance of key host genes, such as Ikaros, in viral transformation and oncogenicity.

## 2. Materials and Methods

### 2.1. Viruses

In the first experiment, which was designed to elicit MD tumors for further genomic characterization, the virulent MDV strain JM/102W (passage 14; [22]) from the USDA, ARS, Avian Disease and Oncology Laboratory (ADOL) stock was used. In the second experiment, which was designed to validate Ikaros as a cancer driver gene, three recombinants MDVs were used. G2M is a recombinant MDV derived from a virulent Md5-strain-based bacterial artificial chromosome (BAC) clone in which an enhanced green fluorescent protein (EGFP) was inserted immediately 3’ of the Meq start codon to allow both EGFP and Meq to be transcribed from the Meq promoter; however, the translated proteins are separated by the 2A self-cleaving peptide [23]. G2M Ikaros WT and G2M Ikaros Mut MDV BACs were generated by modifying the G2M BAC by replacing the EGFP gene in G2M with either the chicken wild-type Ikaros gene (*IKZF1*) or mutant *IKZF1*, respectively. In brief, one repeat long (RL) copy was removed from G2M to produce dRLG2MBAC using the same approach as previously described by Engel et al. [24]; the virulence of the recombinant viruses generated from dRLG2MBAC and G2M BAC were comparable (Appendix A). To replace EGFP with *IKZF1*, two gBlocks with a BamHI site at the 5’ end and a FLAG tag sequence (C terminal end of Ikaros) followed by a Hind III site at the 3’ end were synthesized (Integrated DNA Technologies, Coralville, IA, USA). One gBlock contained the full-length, coding sequence of wild-type Ikaros (WT; GenBank accession no. NM_205088.1). The other gBlock contained a mutant (Mut) *IKZF1* allele found in three MD tumors that was identical in sequence to WT *IKZF1* except for a C to T transition at position 484 resulting in an arginine (R) to cysteine (C) missense. The synthesized fragments were cloned into pBluescript SK+ between the BamHI and HindIII sites and used to replace the EGFP gene in dRLG2M by Red-mediated recombineering [25]. The resultant BAC clones with wild-type and mutated *IKZF1* were named G2M Ikaros WT and G2M Ikaros Mut, respectively (Figure 1).

Purified BAC DNA was used to generate viral stocks by transfecting into chicken embryo fibroblasts and amplified stock prepared after four passages, as described by Hildebrandt et al. [26]. The generated G2M Ikaros WT and G2M Ikaros Mut viruses were confirmed for presence of the intended Ikaros sequence by PCR, followed by Sanger sequencing. Immunocytochemistry was conducted to confirm expression of the introduced Ikaros sequences via immunofluorescence utilizing anti-FLAG antibodies. Specifically, chick embryo fibroblast cultures infected with the G2M Ikaros WT or G2M Ikaros Mut viruses were fixed at 6 days post-infection (dpi) when showing mature MDV plaques with 1:1 acetone:methanol, then stained with a primary anti-FLAG monoclonal antibody clone M2 (Sigma, St. Louis, MO, USA; product no. F1804) and secondary anti-mouse IgG (H+L)-Rhodamine donkey antibody (Life Technologies, Carlsbad, CA, USA; product no. SAB3701099) to visualize confirmation of Ikaros-FLAG expressing plaques by fluorescence (Appendix A).

### 2.2. Bird Experiments and Analyses

All bird experiments were approved by the ADOL Institutional Animal Care and Use Committee; approval no. 13.30 and 2019-02. All chickens were specific pathogen free (SPF) single comb, white leghorns from ADOL pedigreed lines. To generate MD tumors to survey for somatic mutations, highly inbred lines 6_3_ and 7_2_, which are MD-resistant and susceptible, respectively, were intermated to produce F_1_ progeny. At hatch, these maternal antibody negative chicks were challenged intra-abdominally with 1000 plaque-forming units (pfu) of MDV strain JM/102W.

To determine the virulence of recombinant MDVs expressing the *IKZF1* alleles, lines 15I_5_ and 7_1_, both MD-susceptible and maternal-antibody-negative, were intermated to produce F_1_ progeny that were highly MD-susceptible. Progeny were left unchallenged (negative control) or challenged intra-abdominally with 500 pfu of G2M, G2M Ikaros WT, or G2M Ikaros Mut viruses at 5 days of age. Furthermore, in case of early chick mortality, additional birds of the same hatch were kept in a separate isolator to act as replacements. At 6, 13, and 20 dpi, 5 distinct birds in each lot were randomly selected and bled to obtain peripheral blood lymphocytes (PBLs).

Survival curves between lots of infected birds were compared using a log-rank Mantel–Cox test to determine significant differences between viruses (*p* < 0.05). MD incidence was calculated as a percent based on the number MD-positive birds over the total number of birds per lot, and a Fisher’s exact test was used to compare for significant differences between the number of MD-positive birds between groups. Odds-ratio calculations were used to statistically compare tumor incidence between groups of birds based on the number of birds with tumors versus the number of birds without tumors per lot in order to determine risk for tumorgenicity of the Ikaros-expressing viruses relative to G2M.

In all animal trials, when birds became moribund or reached 8 weeks of age, they were euthanized and immediately necropsied.

### 2.3. Tissue Collection and Further Processing

To enhance tumor homogeneity for genomic characterizations, large gross tumors were preferentially collected with the majority being from the gonads. In addition, grossly normal tissue (typically liver) was also collected from the same bird to provide matching controls for DNA. Tissues for DNA sequencing were snap frozen and then stored at −80 °C prior to extraction.

Whole genomic DNA was extracted from frozen tumor and control tissues via the QIAamp DNA Blood Mini Kit (Qiagen, Germantown, MD, USA). DNA integrity and quantity was measured via gel electrophoresis and Qubit (Thermo Fisher Scientific, Waltham, MA USA). All samples underwent DNA sequencing with 125 bp paired-end reads via the Illumina HiSeq 2500 at the Michigan State University Research Technology Support Facility Genomics Core. DNA sequence datasets were deposited in NCBI under accession no. PRJNA767437.

To determine viral loads in PBLs, DNA was extracted and qPCR was conducted as described by Dunn and Silva [27]. In brief, MDV and chicken DNA levels were determined by comparison to seven 10-fold serial dilution standards of UL27 (glycoprotein B aka gB) and GADPH, respectively. Relative viral load was the gB-to-GADPH ratio.

### 2.4. Bioinformatics

An extensive bioinformatics pipeline was developed for this project and more. It can be accessed at two GitHub repositories: https://github.com/hongenxu/MDV_proj and https://github.com/steepale/IKZF1_paper_code (both accessed on 22 February 2016).

#### 2.4.1. DNA Sequencing

The following analysis was designed following the Genome Analysis Toolkit (GATK) best-practices pipeline [28]. Reads were inspected for quantity and quality before and after trimming with FastQC (v.0.11.3) [29]. Reads were trimmed of low-quality bases and primers via Cutadapt (v.1.14) [30]. Trimmed reads were aligned to the Gallus_gallus-5.0 reference genome with BWA-MEM [31]. Read-group annotation was added via Picard tools (v.1.113) [32]. Reads within each sample and sequencing lane were filtered of duplicate reads and were realigned around indels via Picard tools (v.1.113) and GATK (v.3.7.0) [33]. Indel realignment was performed once more after reads of the samples were merged. Additional processing procedures were performed with SAMtools (v.1.3.1) [34,35].

#### 2.4.2. Detection of Somatic Single-Nucleotide Variants (SNVs) and Small Insertions and Deletions (Indels)

Somatic SNVs and indels were called from whole genome sequencing data using six and four callers, respectively. Somatic SNV callers included MuSE (v1.0rc_c039ffa) [36], MuTect2 (v1.1.7) [37], JointSNVMix2 (v0.7.5) [38], SomaticSniper (v1.0.5.0) [39], VarDict (v1.4.4) [40], and VarScan2 (v2.4.1) [41]. Somatic indels were called using VarScan2 (v2.4.1) [41], MuTect2 (v1.1.7) [37], JointSNVMix2 (v0.7.5) [38], and VarDict (v1.4.4) [40]. The default hard filters were used for all algorithms. For algorithms suspected of generating a high frequency of false positives in their raw outputs (e.g., JointSNVMix2 and VarDict), we incorporated an additional hard-filtering step before their outputs were compared to other callers.

Somatic SNVs and indels were further filtered. Genomic loci with reads of mapping and base qualities ≥20 were queried for in their respective BAM files with SAMtools (v1.3.1) [34,35]. High-quality reads were required to demonstrate a variant allele frequency ≥0.05 (in tumor sample reads) and coverage of at least 4× (in tumor and match normal sample reads). Each putative somatic variant was queried within tumor BAM files manually with the Integrative Genomics Viewer (IGV; v2.3.91) [42,43] and with a custom script using SAMtools mpileup (v1.3.1) [34,35].

#### 2.4.3. Annotation of IKZF1 Variants on Ikaros Protein Domains

Ikaros protein isoforms were collected from the UniProt database [44] and referenced from IKZF1-201 and IKZF1-202 transcript sequences from both Ensembl [45] and RefSeq [46]; see Appendix A for a view of the *IKZF1* in the chicken genome, as well as the known isoforms. Ikaros isoforms were compared to proteins in the UniRef90 database [47] via pBlast (E-threshold of 0.001) [48,49], and resulting protein sequences were aligned via Clustal Omega (v1.2.4) [50]. Hierarchical analysis of amino acid residue conservation [51] was performed within JalView version 2 [52] and results were reduced to 5 species (Appendix A).

## 3. Results

### 3.1. MD Tumors Contain Key Somatic Mutations in IKZF1

To identify somatic mutations in MD tumors, 200 line 6_3_ × 7_2_ F_1_ birds were challenged with virulent MDV to generate tumors, and the largest-sized tumor samples from 22 birds were selected and further characterized. The most frequent and recurrent somatic mutations across tumors occurred in *IKZF1*, the gene encoding the transcription factor Ikaros. Of the MD tumors tested, 9 of 22 contained somatic nonsynonymous mutations in *IKZF1*, the only gene to demonstrate enrichment for nonsynonymous somatic mutations and diverse mutation types across tumors.

In total, 10 unique somatic nonsynonymous *IKZF1* mutations were found across the nine tumors (Table 1) with all somatic variants clustered in two critical N-terminal C_2_H_2_ zinc-finger binding domains (Figure 2). Ikaros isoforms 1 (Ensembl IKZF1-201; RefSeq IKZF1-X1) and 2 (Ensembl IKZF1-202; RefSeq IKZF1-X3) were chosen for this analysis because chicken Ikaros isoforms 1 and 2 closely resemble the human Ikaros isoforms 1 and 2, with 86% and 78% amino acid identity, respectively; and isoforms 1 and 2 are the most abundantly expressed throughout development of hematopoietic cells in both human and mouse.

### 3.2. Validation That Mutant Ikaros Allele Promotes MDV-Induced Transformation

To validate that *IKZF1* is an MD cancer driver gene, highly MD-susceptible chickens were either left unchallenged or challenged with various recombinant MDVs. Two replicate cohorts with four experimental groups each were measured for MD and tumor incidence: one group (10 chicks) served as a control with challenge-free birds, and three groups (at least 15 chicks per treatment) were challenged with either G2M, G2M Ikaros WT, or G2M Ikaros Mut. The results of both replicates are shown in Table 2; differences in the number of birds between treatment groups were due to early chick mortalities that were unrelated to MD.

As shown, there were large differences in both MD and tumor incidence based on whether a bird was challenged with virulent MDV and, when applicable, the Ikaros allele expressed. The key finding was that expression of either the WT or Mut Ikaros allele had a significant impact with birds infected with the G2M Ikaros WT yielding low (~12%) disease or tumors, while in stark contrast, birds infected with G2M Ikaros Mut had high (~90%) disease and tumor incidence. When comparing MD incidence between the birds challenged by Ikaros WT and Ikaros Mut, there were highly significant differences in MD incidence for both replicates 1 and 2 depending on infection with the different Ikaros alleles (*p* < 0.0001 for both replicates). Furthermore, birds infected with G2M and G2M Ikaros Mut demonstrated more tumors internally on average, as revealed by counts of organs harboring tumors (Appendix A). Furthermore, all tumors in birds infected with G2M Ikaros WT were small and required further verification. Particularly for birds infected with G2M Ikaros Mut, these cohorts had significantly higher probability of developing tumors relative to G2M (Replicate 1 odds ratio of 33, *p* = 0.0013; Replicate 2 odds ratio of 24, *p* = 0.0053) as opposed to the birds infected with G2M Ikaros WT, which did not have as large of an increased risk for tumor development relative to G2M controls (Replicate 1 odds ratio 0.92, *p* = 0.94; Replicate 2 odds ratio 0.20, *p* = 0.071). These results were consistent across both replicates, and strongly suggested that the addition of mutant *IKZF1*, in the context of MDV, further drives MD onset and tumor formation.

### 3.3. Lifespan of Birds in the G2M Treatment Group Were Significantly Shorter Than Other Groups

Survival of birds over time (Figure 3) showed significant differences for both G2M Ikaros WT or G2M Ikaros Mut viruses relative to the parental G2M virus lacking any Ikaros allele. In both the Ikaros WT or G2M Ikaros Mut groups, the majority of birds survived the duration of the experiment, leading to a significant difference in the survival curves of the two recombinant G2M viruses relative to parental G2M, but nonsignificant differences in survival of the two Ikaros viruses relative to each other (Ikaros-expressing viruses vs. G2M, *p* < 0.0001 for both replicates; G2M Ikaros Mut vs. G2M Ikaros WT, *p* = 0.32 and *p* = 0.26 for Replicates 1 and 2, respectively).

Traditionally, MD has been defined by the presence of either gross visceral tumors or enlarged nerves. Since these phenotypes can only be determined at necropsy, the correlation of survival and MD is not high, as birds can often develop one or more tumors but still live to the end of the experiment. To explore whether this correlation could be improved, the criteria for MD incidence was expanded to include birds at least 7 days of age that died or developed clinical disease, and had bursa or thymic atrophy (BTA) with a score of 3 or higher (0 to 4 scale). The justification for this expanded definition was that MDV typically leads to BTA due to replication in these organs soon after infection; thus, birds that died early with no tumors or nerve enlargement but had BTA likely succumbed to MD. Comparison of survival and MD incidence, both the traditional and expanded definitions, over time are shown in Appendix A. As shown, the expanded definition of MD improved the relationship between length of survival and MD.

### 3.4. In Vivo Replication and Viremia of Recombinant MDVs Expressing Ikaros Alleles

Considering the lack of disease with the G2M Ikaros WT, it would be reasonable to hypothesize that this deficiency was due to this virus replicating poorly or to significantly less levels compared to those MDVs that did induce tumors. To test this hypothesis, qPCR data showed that the G2M virus had high levels of viral replication, as expected for a virulent MDV strain (Figure 4). Conversely, both G2M Ikaros WT or G2M Ikaros Mut viruses had comparable levels of MDV replication to each other, but both replicated at lower levels compared to parental G2M virus. Therefore, the differences in MD incidence and tumor formation observed between G2M Ikaros WT or G2M Ikaros Mut viruses could not be attributed solely to deficiencies in replication.

## 4. Discussion

With the widespread usage of Rispens as an MD vaccine in commercial flocks, MD incidence has steadily declined. However, due to the repeated history of more virulent MDV strains, MD is still considered a threat to the poultry industry. For this reason and to pursue fundamental knowledge to further leverage existing efforts, our group has been practicing a strategy of identifying chickens with enhanced genetic resistance to MD based on genomic selection (GS) as an alternative and/or sustainable method for MD control. Specifically, our goal is to identify genetic markers or, even better, causative polymorphisms that are associated with MD resistance for use in GS. GS is now widely practiced in commercial poultry breeding, as it offers many advantages over traditional phenotypic selection. For genetic improvement of disease resistance to pathogens such as MDV, GS is highly advantageous, as there is no need to expose elite flocks to a hazardous pathogen. Furthermore, one can readily select birds of both sexes and at an early age.

Our work utilizing allele-specific expression (ASE) in response to MDV challenge was extremely successful in identifying genetic markers that proved useful in GS [53]. We hypothesized that the major mechanism underlying the complex trait of MD genetic resistance was variation in gene expression. Using ADOL lines 6_3_ (MD-resistant) and ADOL line 7_2_ (MD-susceptible), we were able to validate this hypothesis. More importantly, we were able to show that ASE single-nucleotide polymorphisms (SNPs) were highly accurate in GS, and these genetic markers could account for a remarkable 83% of the observed genetic variance. Efforts are currently underway to find the causative SNPs, which are likely in regulator elements (i.e., promoters and enhancers). Functional annotation of the chicken genome using materials including lymphocytes from line 6_3_ × 7_2_ F_1_ birds [54] greatly increases the power of this approach.

Another parallel approach to identify actual causative alleles for GS is based on the fact that genetically resistant MD birds are characterized by the lack of lymphomas (phenotype). Thus, similar to the human cancer field, a recent focus by our group has been to identify genes with specific alleles (genotype) that drive tumor formation. In other words, we wish to identify all the MD cancer driver genes, which would also define the genomic landscape of MD tumors. These genes, once found, besides providing precise alleles that are less likely to incur somatic mutations and corresponding causative markers for genetic improvement, should provide basic biological information that we believe can be leveraged further by the field. In the work reported here, we identified and validated Ikaros (*IKZF1*) as the first MD cancer driver gene.

Based primarily on knowledge gained from human and mouse studies, Ikaros (*IKZF1*) and its four other related family members (Helios (*IKFZ2*), Aiolos (*IKFZ3*), Eos (*IKFZ4*), and Pegasus (*IKFZ5*)) are considered the master regulators of lymphocyte development. They do so by encoding Zn-finger transcription factors that regulate both gene expression and chromatin remodeling of lymphoid cells. For example, binding of Ikaros to the CD8alpha gene locus promotes thymocytes to differentiate more toward CD8 cells vs. CD4 [55]; MD tumors are predominantly transformed CD4 cells. Ikaros and its family members also play a prominent role in the differentiation of effector CD4+ T-helper-cell subsets [56]. There is very limited information on Ikaros function in avians. Based on conserved evolutionary history in vertebrates [57] and a study on Ikaros regulation in chicken B cells [58], it is highly likely that Ikaros functions in a similar manner in chickens.

*IKFZ1* is a known tumor-suppressor gene, and has been shown to be frequently mutated in human B-cell acute lymphoblastic leukemia (B-ALL) and diffuse large B-cell lymphoma (DLBCL). The majority of the somatic mutations are found in the middle two N-terminal Zn-finger binding domains, which disables the ability of the protein to attach to DNA, thereby destroying its function [59]. Mice bioengineered with Ikaros knockouts or key somatic mutations such as those in the two key Zn-finger DNA binding domains develop leukemias and lymphomas, yet potential roles of Ikaros in oncogenesis during viral transformation were less clear [60,61,62,63].

We also saw clustering of highly deleterious mutations in specific N-terminal Zn-finger binding domains, which is the same pattern observed in human ALL [64]. More importantly, all of the observed MD somatic mutations are in highly conserved amino acids (Appendix A). Specifically, all the nonsynonymous mutations are conserved across species from sea lamprey to human, the two most distant species with orthologous Ikaros [65]. Furthermore, all residues targeted by missense mutations and in-frame deletions are considered essential for human Ikaros to bind to DNA [66].

We also see hemizygous mutations; i.e., only one allele with a somatic mutation (data not shown). Thus, it is likely that the mutant Ikaros protein acts in a dominant negative fashion, which has also been observed in humans [67]. This is because the C-terminal Zn-finger domains, which are not altered, can still enable Ikaros to produce homodimers or heterodimers with other Ikaros family members, but due to the mutant N-terminal Zn-finger domains, the dimer cannot properly bind to DNA. Unfortunately, at least for *IKZF1,* because of this dominant negative action, we could not achieve our major goal of identifying an MD-resistant allele for use in GS. Ongoing efforts have identified a number of other candidate MD cancer drive genes (manuscript in preparation) that may be more productive for GS. It should be noted that despite concerted efforts, we did not find any somatic mutations in p53, as was previously reported by Zhang and coworkers [68]. This discrepancy could be due to a difference in the MDV strain used or the genetics of the birds.

Nevertheless, our experiments assessing the role of recombinant MDV showed the significant role of WT Ikaros in reducing tumor formation despite the presence and expression of Meq, the viral oncogene. Alternatively, infection with recombinant MDV containing both the Meq oncogene and the mutated form of Ikaros, predicted to disrupt the tumor-suppressor function, resulted in a significantly greater percentage of MD incidence and a significantly higher risk for tumor formation relative to both parental and WT Ikaros MDVs. This revealed that the known dominant-negative impact of somatic Ikaros mutations leading to oncogenesis is also applicable in viral transformation models of tumorigenesis.

A question that needs to be addressed in the future is: how are the initial *IKZF1* somatic mutations acquired? There are two possible explanations that relate somatic mutation rate with MD genetic resistance. First, MD-resistant birds may have lower somatic mutations rates compared to MD-susceptible birds. Second, as MD-susceptible birds have higher MDV viremia levels, it is likely that more CD4 lymphocytes are activated, thereby increasing the likelihood that one or more cells during replication will acquire a key *IKZF1* somatic mutation and be infected by MDV.

The discovery of Ikaros as an MD cancer driver gene helped to resolve the mystery as to why virulent MDV is not sufficient to induce tumors in every infected bird. Levy et al. [69] reported that Meq was only weakly oncogenic. Thus, it is more likely that Meq inhibits apoptosis, possibly through interaction with Bcl-2 [70] and p53 [71], and combined with somatic mutations in *IKZF1* or other MD cancer driver genes, leads to the formation of gross tumors.

Ikaros is also a major factor in the maintenance of viral latency in Epstein–Barr virus (EBV), a related herpesvirus [72]. Ikaros does this by altering the expression of Oct-2, Bcl-6, and other transcription factors that direct EBV reactivation and plasma cell differentiation. Based on this information, it would be interesting to speculate that Ikaros is also involved in influencing MDV latency, which may also help explain why both our recombinant MDVs expressing Ikaros had low replication levels compared to the G2M parental virus. Expression of Ikaros also helps to explain the role of EBV as a human-tumor-associated virus, since viral gene EBNA-1 is necessary for B-cell transformation, and is required for replication and maintenance of EBV episomes during latency.

## 5. Conclusions

MDV-induced transformation requires at least two hits: (1) additional somatic mutations in key chicken cancer genes such as *IKZF1* that drive unregulated cellular growth, and (2) Meq to inhibit apoptosis. Therefore, common mechanisms for Ikaros mutations functioning as a cancer driver gene for tumorigenesis apply in both somatic and viral transformation oncogenesis models.

## Figures and Tables

**Figure 1 microorganisms-10-00401-f001:**
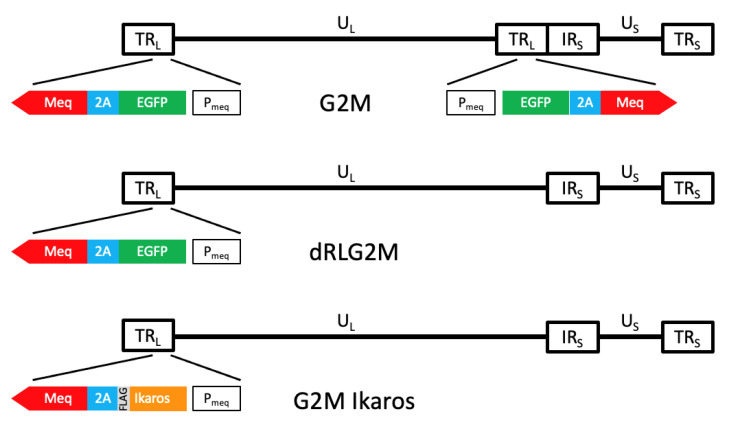
A schematic presentation of the recombinant MDVs. G2M is the base infectious BAC clone containing the entire Md5 strain MDV genome with EGFP inserted immediately 3’ of the Meq start codon and separated from Meq by a 2A self-cleaving peptide. Next, dRLG2M was generated by removing one repeat long (RL) copy. Finally, G2M Ikaros was generated by replacing EGFP with either the wild-type or mutant *IKZF1* allele.

**Figure 2 microorganisms-10-00401-f002:**
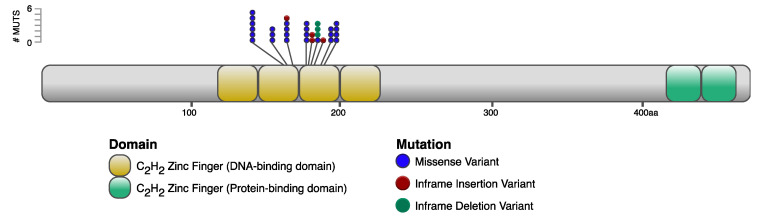
A schematic presentation of the Ikaros protein showing the collection of somatic nonsynonymous mutations in MD tumors that cluster on essential amino acids in N-terminal zinc fingers 2 and 3 (DNA-binding).

**Figure 3 microorganisms-10-00401-f003:**
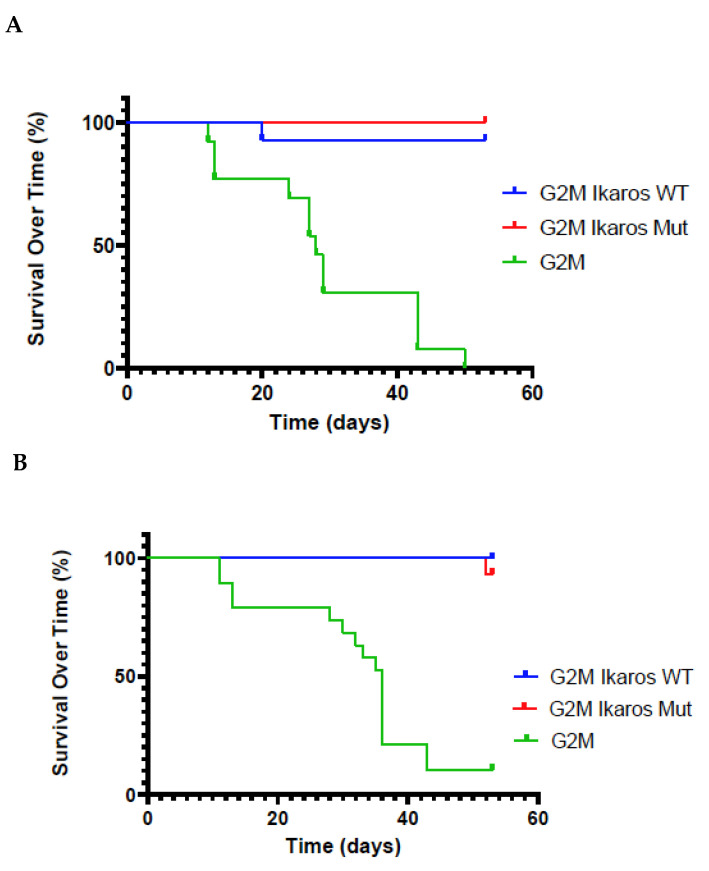
Survival curves of chickens in replicates across experimental groups. Chick mortalities that occurred prior to 10 days of age were excluded from analysis in both Replicate 1 (**A**) and Replicate 2 (**B**).

**Figure 4 microorganisms-10-00401-f004:**
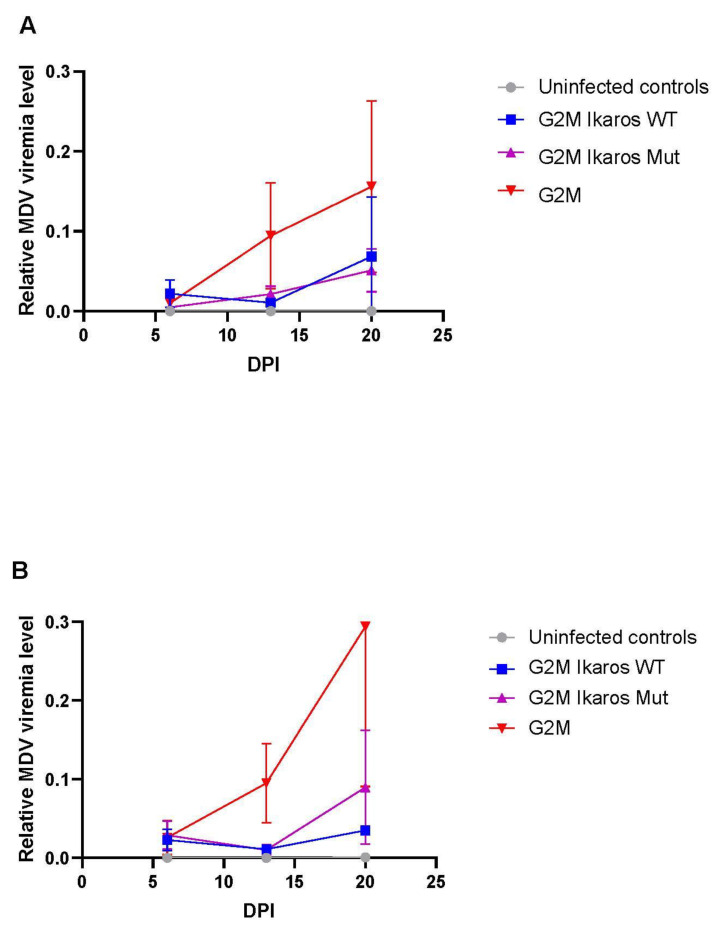
*In vivo* replication of recombinant viruses across treatment groups based on qPCR of PBLs (*n* = 5 birds per timepoint) for both Replicate 1 (**A**) and Replicate 2 (**B**). MDV viremia levels are the ratio of MDV *UL27* (glycoprotein B) to *GADPH* levels.

**Table 1 microorganisms-10-00401-t001:** Somatic mutations in the DNA-binding domain of *IKZF1*.

Position ^1^	Reference ^2^	Alternative ^3^	Variant Type	AA Change	Sample ^4^	Prediction
80,972,101	C	T	missense	Arg162Cys	777, 851, 901	deleterious
80,972,102	G	T	missense	Arg162Leu	835	deleterious
80,972,104	C	T	missense	His163Tyr	842, 927	deleterious
80,972,116	C	T	missense	His167Tyr	901	deleterious
80,972,118	C	G	missense	His167Gln	756	deleterious
80,972,141	G	A	missense	Cys175Tyr	901	deleterious
80,972,141	G	GCCA	inframe insertion	His176dup	901	deleterious
80,972,149	TGTAACTACGCCTGCCGGCGCA	T	inframe deletion	Cys178 to Arg185 delinsTrp	911	deleterious
80,972,152	A	AACT	inframe insertion	Tyr180dup	918	deleterious
80,972,167	C	T	missense	Arg184Cys	927	deleterious

^1^ Position on chr. 2 based on the Gallus_gallus-5.0 reference; ^2^ wild-type allele; ^3^ mutant allele; ^4^ last 3 digits of wingband used to identify the bird.

**Table 2 microorganisms-10-00401-t002:** Tumor incidence in control birds or those infected with various recombinant MDVs.

Replicate	Treatment	Total Birds	MD ^1^		Tumor Positive	
			Count	Percent	Count	Percent
1	none	6	0	0	0	0
	G2M	13	4	31	2	15
	G2M Ikaros WT	14	2	14	2	14
	G2M Ikaros Mut	14	13	93	12	86
2	none	10	0	0	0	0
	G2M	19	9	47	7	37
	G2M Ikaros WT	19	2	11	2	11
	G2M Ikaros Mut	15	14	93	14	93

^1^ MD was considered positive when a bird had enlarged nerves or evidence of a tumor.

## Data Availability

All raw sequencing data analyzed in this study were submitted to the NCBI Sequence Read Archive under the BioProject PRJNA767188.

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
