# Peer review of "Identification and Validation of Ikaros (IKZF1) as a Cancer Driver Gene for Marek’s Disease Virus-Induced Lymphomas"

_microorganisms, 2022, doi:10.3390/microorganisms10020401_

Round 1
Reviewer 1 Report
General Comments
Strengths: The science is sound, and the manuscript is well written.
Weaknesses: Figure 3 is not easy to follow, may be explain by delta-delta Ct method. In addition, brief illustration of the IKAROS gene in the introduction or elsewhere, may be helpful for the readers to understand about the gene.
Other comments: What other IKAROS isoforms are present in chickens? Do they play similar role. Please include more information on what is known and what remains unknown in the introduction and/or discussion.
Specific comments
Figure 3: Is it possible to explain this figure by delta-delta Ct method?
Reviewer 2 Report
Summary: In this research article by Steep et al, somatic mutations in the Ikaros gene was identified in tumors of chickens infected with Marek’s disease virus (MDV). To do this, highly susceptible chickens were infected with a virulent strain of MDV called JM/102W. Using next generation sequencing of tumors, the authors found somatic mutations within the chicken Ikaros gene in 9 of 22 tumors. Most of these mutations were located within the zinc finger DNA binding domain of Ikaros. The authors postulated that these mutations drive tumorigenesis in chickens. They then proceeded to develop recombinant viruses in which either wild type or mutant Ikaros proteins are expressed in the context of the virus. Here, they use the recombinant Md5 strain that is generally considered very virulent. Following inoculation into chickens in two experiments, they found viruses infected with the mutant Ikaros expressing MDV had a very high MD incidence (93%) compared to viruses expressing no or wild-type Ikaros (<47%). These results are very intriguing; however, the manuscript has significant problems making their interpretation of the data difficult.
Broad Comments: The article is well written but contains more information than is needed for data shown. The finding that Ikaros mutations were found in 9 of 22 tumors is intriguing, but this is merely suggestive. Unfortunately, the data using rMDV does not fully support their hypothesis since the data seems confusing. Specific comments are below.
Major Specific Comments:
- A major critique of the data is related to the viruses used in the study. Importantly, when infecting highly susceptible chickens with the parental (G2M) virus, <50% of chickens developed MD (Table 2) suggesting the rMDV used are defective. This has been shown previously in Tai et al (2017). Although the strategy of replacing GFP with the Ikaros WT and Mut genes is innovative, the replication of the rMDVs (Fig. 3) and Ikaros expression (Supp. Fig. 1) suggests this approach is flawed. Although, the MD incidence data (Table 2) suggests large differences between the groups, the survival curves (Fig. 2) contradict this (see below). Also, the authors lack proper controls to show the viruses are providing the expression needed. Supplementary Figure 1 is not conclusive. It is difficult to ascertain whether the low level of fluorescence seen is positive for only autofluorescence. Proper controls need to be included.
- The data in Table 2 and the survival curves (Fig. 2) are confusing. The authors show only 31 and 47% of chickens developed MD in the G2M groups (Table 2), but survival is down to 0%. What did they die of?While G2M Ikaros Mut has 93% MD incidence in both replicates, yet the chickens have over 90% survival. Why didn't they die? This needs to be resolved or discussed.
- It would be nice to have a schematic representation of the rMDVs used in this study. It is confusing when reading the description (lines 117-139). No in vitro characterization was shown. Additionally, the data on the virulence of the dRLG2MBAC viruses should be included as the virulence is important for these studies. Also, was the miniF sequence removed from the viral genome following reconstitution, since it is well known having the sequence present attenuates MDV (Zhao et al, JVI 2008, Jarosinski et al, JVI 2007)
- The introduction is exceedingly long for this work. Some subjects that can be eliminated is the lengthy introduction on MD vaccines (lines 50-69) and MDV integration (lines 81-93). Neither of these topics are discussed and can be mentioned, but paragraphs devoted to the subject matter is not warranted.
Author Response
Please see the attachment.
(We apologize for the length of time on this revision. Due to ARS pandemic restrictions (25% max capacity) and the move of laboratory facilities in Athens, GA, we were unable to perform the needed lab work for the revised Supplementary Figure 1 until recently.)

Round 2
Reviewer 2 Report
I am satisfied with most of the changes to the manuscript. However, I have to issues still.
First, the article calls Ikaros a cancer “driver” but the data and results show it is actually a cancer "suppressor." That would also fall in line with the literature showing Ikaros is a “suppressor”. A better terminology should be developed because it is confusing to call a cellular tumor suppressor gene a cancer “driver.” For example, I have never heard of p53 called a cancer "driver" when mutated. I understand what the authors are getting at, but it would better to use a term that is specific for this proteins function. Meq could be considered a cancer “driver.”
Secondly, with respect to the comment. "Thus, our results are not a contradiction and a clear representation of the actual results."
Thank you for your explanation, although I am aware of the differences between survival times and MD incidence. The major confusion comes from how the data is presented as the survival curve is a figure showing survival over time, while MD incidence is in Table form that does not represent data over “time.” The data are not comparable without considering the time of MD incidence determination. When “MD” is diagnosed is not shown in the table, but is a major point of the data, correct? Were all the tumors found at 8 weeks of age? That is what the Table suggests without any explanation. Thus, the confusion is in the data presentation. I would suggest an MD incidence figure, like the survival time figure, to better show your point that though the Ikaros Mut group birds survive, at termination they had significant numbers of tumors (MD). The G2M group apparently died throughout the experiment with no birds left after <50 days. But only 31 and 47% had MD (based on tumors or enlarged nerves). Did they have any signs? MD is more than just tumors and enlarged nerves including wasting, neurological symptoms, and metabolic disease (see Osterrieder 2006). Alternatively, show tumor incidence and not MD since the point is that tumor induction is significantly increased with mutant Ikaros, correct? In all, the authors should strive to present the data so a general audience can understand the data, without knowing the nuances of MD research. Currently, without any explanation, it is confusing.
The article calls Ikaros a cancer “driver” but the data and results show it is actually a suppressor. That would also fall in line with the literature showing Ikaros is a “suppressor”. A better terminology should be developed because it is confusing to call a cellular tumor suppressor gene a cancer “driver.”
